# Bisphenol A (BPA) Directly Activates the G Protein-Coupled Estrogen Receptor 1 and Triggers the Metabolic Disruption in the Gonadal Tissue of *Apostichopus japonicus*

**DOI:** 10.3390/biology12060798

**Published:** 2023-05-31

**Authors:** Jieyi Yuan, Jingwen Yang, Xiuwen Xu, Zexianghua Wang, Zhijing Jiang, Zhiqing Ye, Yucheng Ren, Qing Wang, Tianming Wang

**Affiliations:** 1National Engineering Research Center of Marine Facilities Aquaculture, Marine Science and Technology College, Zhejiang Ocean University, Zhoushan 316022, Chinasilence84309@163.com (J.Y.);; 2Research and Development Center for Efficient Utilization of Coastal Bioresources, Yantai Institute of Coastal Zone Research, Chinese Academy of Sciences, Yantai 264003, China

**Keywords:** *Apostichopus japonicus*, reproduction, estradiol receptor, Bisphenol A (BPA), metabolic disruption

## Abstract

**Simple Summary:**

*Apostichopus japonicus* is a temperate marine invertebrate and an economically important aquatic echinoderm species in China. As a benthic organism, sea cucumbers feed on small benthic particulate matter and are easily affected by pollutants. Therefore, for conservation purposes, it is important to elucidate the effects of specific pollutants on these animals. Bisphenol A (BPA) has been identified as an endocrine disruptor that functions as an estrogen analog and typically causes reproductive toxicity by interfering with the endocrine system. We identified the G protein-coupled estrogen receptor 1 in *A. japonicus* and found that both BPA and estradiol could activate the estrogen-receptor-mediated mitogen-activated protein kinase signaling pathway. Using biological approaches, we have confirmed the estrogen-analog bioactivity of BPA on sea cucumbers by direct interaction with the estrogen receptor. We also determined that BPA affects the reproduction of sea cucumbers by metabolic disruption in ovary tissue.

**Abstract:**

The sea cucumber, *Apostichopus japonicus,* is a marine benthic organism that feeds on small benthic particulate matter and is easily affected by pollutants. Bisphenol A (BPA, 4,4′-isopropylidenediphenol) has been identified as an endocrine disruptor. It is ubiquitously detectable in oceans and affects a variety of marine animals. It functions as an estrogen analog and typically causes reproductive toxicity by interfering with the endocrine system. To comparatively analyze the reproductive effects of estradiol (E_2_) and BPA on sea cucumbers, we identified a G protein-coupled estrogen receptor 1 (GPER1) in *A. japonicus* and investigated its effects on reproduction. The results showed that BPA and E_2_ exposure activated *A. japonicus* AjGPER1, thereby mediating the mitogen-activated protein kinase signaling pathways. High-level expression of *AjGPER1* in the ovarian tissue was confirmed by qPCR. Furthermore, metabolic changes were induced by 100 nM (22.83 μg/L) BPA exposure in the ovarian tissue, leading to a notable increase in the activities of trehalase and phosphofructokinase. Overall, our findings suggest that AjGPER1 is directly activated by BPA and affects sea cucumber reproduction by disrupting ovarian tissue metabolism, suggesting that marine pollutants pose a threat to the conservation of sea cucumber resources.

## 1. Introduction

The sea cucumber, *Apostichopus japonicus* (Selenka), is a temperate marine invertebrate and an economically important aquatic echinodermic species in China [1,2]. As benthic organisms, sea cucumbers feed on small benthic particulate matter and are easily affected by pollutants [3]. Thus, it is important to investigate the effects of pollutants on sea cucumbers to conserve wild sea cucumber species. According to estimates, 5–10% of all plastic products produced annually are transported to the ocean, exposing marine organisms to bisphenols produced by plastic debris in the marine environment [4,5]. Bisphenol A (BPA, 4,4′-isopropylidenediphenol) is a xenoestrogen or endocrine-disrupting chemical (EDC) mainly used in the production of polycarbonate and epoxy resins [6]. It is widely utilized in food-contact materials, such as food-packaging bags, bottles, and coatings, and is present in various concentrations in the atmosphere, dust, water, and food [7,8,9]. The average BPA concentrations in seawater from the coastal areas of China are 449.2/186.3 ng/L in winter and summer, respectively, whereas sediment BPA concentrations in the North Bay of the South China Sea range from 0.56 to 5.22 ng/g dry weight (DW) [10,11]. In addition, as a xenoestrogen, BPA can significantly damage the reproductive, immune, and neuroendocrine systems of marine organisms [12,13,14].

Estrogen belongs to a family of female hormones including estrone (E_1_), estradiol-17β (E_2_), estriol (E_3_), and estretrol (E_4_), which mediate a wide range of physiological functions via estrogen receptors [15]. BPA induces endocrine-disrupting effects by acting as an estrogen-receptor agonist [16]. It can bind to a variety of receptors, including estrogen receptors, interfere with diverse endocrine factors, and negatively affect biological functions [17,18]. Recent studies have revealed that exposure to BPA directly harms aquatic species by causing oxidative damage and metabolic disruption [19,20,21]. Additionally, exposure to BPA significantly increases neurotoxicity and immunotoxicity in marine invertebrate blood clams [14]. Although numerous invertebrates play a significant role in marine ecological systems and constitute the largest population of marine organisms, the potential molecular toxicity mechanisms of BPA in these organisms are still not well understood.

G protein-coupled estrogen receptor 1 (GPER1), also known as G protein-coupled receptor 30 (GPR30), is a seven-transmembrane G protein-coupled receptor (GPCR). In recent years, it has been extensively studied as a potential mediator of estrogen-induced physiological changes in vertebrates for its neuroprotective effect by inhibiting toll-like receptor 4 (TLR4)-mediated microglial inflammation and its involvement in the control of zebrafish follicular development by regulating vitellogenesis and epidermal growth factor receptor (EGFR) expression [22,23]. In addition, increasing evidence has shown that GPER1 is crucial for metabolic regulation in vertebrates [24]. GPER1 is activated by estrogen and mediates various signaling pathways, including the cyclic adenosine monophosphate (cAMP)/protein kinase A (PKA) and phospholipase C (PLC)/protein kinase C (PKC) pathways [15,25,26]. BPA has been shown to act as a GPER1 agonist, activating the receptor to trigger intracellular Ca^2+^ influx and PKC in mammals and lower vertebrates [17,27]. However, to the best of our knowledge, no study has investigated the molecular mechanisms underlying endocrine disruption by BPA in invertebrates.

To evaluate the potential effects of BPA on sea cucumbers, we cloned *A. japonicus* GPER1 (AjGPER1), measured the direct interaction between BPA and AjGPER1 via cell signal transduction, and investigated the reproductive toxicity of BPA in gonadal tissues in vitro. This study elucidates the molecular activity of BPA on GPER1 in echinoderms, demonstrates the potential regulatory mechanism of BPA reproductive toxicity in benthic invertebrates, and provides a reference for the conservation of wild sea cucumber species.

## 2. Materials and Methods

### 2.1. Sequence Characterization and Phylogenetic Analysis

AjGPER1 was identified from the *A. japonicus* genome [28] using the Basic Local Alignment Search Tool (BLAST; http://blast.ncbi.nlm.nih.gov/, accessed on 10 March 2020) (accession: PRJNA413998) against GPER1 protein sequences from vertebrate species (*Homo sapiens*, *Mus musculus*, *Canis lupus familiaris*, *Gallus gallus*, *Danio rerio*, *Carassius auratus*, *Larimichthys crocea*, and *Xenopus laevis*). Protein transmembrane helices of AjGPER1 were predicted using TMHMM 2.0 (https://services.healthtech.dtu.dk/service.php?TMHMM-2.0, accessed on 3 December 2021) and their physicochemical properties were predicted using ProtParam (https://web.expasy.org/protparam/, accessed on 5 December 2021) [29,30]. The N-linked glycosylation and phosphorylation sites of AjGPER1 were predicted using the NetNGlyc 1.0 (https://services.healthtech.dtu.dk/service.php?NetNGlyc-1.0, accessed on 7 December 2021) and NetPhos 3.1 servers (https://services.healthtech.dtu.dk/service.php?NetPhos-3.1, accessed on 7 December 2021), respectively [31]. Visualization of the protein morphology and interactive integration of annotation prediction sequence features were performed using Protter 1.0 (http://wlab.ethz.ch/protter/start/, accessed on 9 May 2022). The 3D structure was predicted using the Robetta software (https://robetta.bakerlab.org/, accessed on 5 May 2022) [32]. Multiple sequence alignments of putative AjGPER1 amino acid sequences were conducted using ClustalW, and a phylogenetic tree was constructed based on the maximum likelihood (ML) phylogenetic reconstruction algorithm with 1000 bootstrap trials using MEGA 5.0. The amino acid sequences and corresponding GenBank accession numbers used for multiple sequence alignment and phylogenetic tree analysis are listed in Appendix A.

### 2.2. Sample Collection and cDNA Preparation

Sea cucumber *A. japonicus* individuals (weight: 76.2 ± 5.2 g) were collected from the Weihai Rushan breeding base of the Chinese Academy of Sciences (Weihai, Shandong, China) for cDNA cloning, analyzing gene expression distribution characteristics, and tissue culture. Each batch was acclimated in seawater aquaria (salinity range: 31.38–33.53 ppt), maintained at a consistent temperature (16.0 ± 0.3 °C), and fed with sea cucumber fermented pellet feed (Weihai Yuyang Biotechnology, Weihai, China) for 7 days. The respiratory tree, nerve ring, intestine, muscle, tentacle, and ovarian tissues were collected for further processing. The remaining tissue samples were immediately frozen and stored in liquid nitrogen.

The collected tissue samples of *A. japonicus* were immediately homogenized in TRIzol reagent (TaKaRa, Kusatsu, Japan) and phenol-chloroform (Sinopharm, Shanghai, China). Total RNA completeness was confirmed by agarose gel electrophoresis, and the concentration and quality of the RNA were determined using Nanodrop 2000 (Thermo Fisher Scientific, Waltham, MA, USA). Reverse transcription and rapid amplification of cDNA from each sample were conducted using a mixture of 1 μg total RNA and Oligo-dT (Sangon Biotech, Shanghai, China) heated to 70 °C for 15 min. The Moloney Murine Leukemia Virus (M-MLV) reverse transcriptase, 5 × M-MLV buffer, and recombinant RNase inhibitor (used for RNA degeneration inhibition during cDNA synthesis) (TaKaRa, Kusatsu, Japan) were used according to the manufacturer’s instructions to ensure reverse transcription into single-stranded cDNA. The cDNA was stored at −80 °C in a freezer for subsequent experiments.

### 2.3. Plasmid Construction

To obtain the coding sequence (CDS) of *AjGPER1* by pyrosequencing, forward (*AjGPER1-F*) and reverse primers (*AjGPER1-R*) (Appendix A) were synthesized (Sangon Biotech, Shanghai, China). To explore the localization of AjGPER1 in human embryonic kidney 293 (HEK293) cells (estrogen-receptor negative), the receptor and enhanced green fluorescent protein (EGFP) recombinant expression vector (AjGPER1-EGFP) was constructed and synthesized using the pEGFP-N1 plasmid by the Aoqian Biotechnology Company (Hangzhou, China). Furthermore, both the human endoplasmic reticulum resident protein CALR (NM_00434.3) and Golgi resident protein TGN38 (NM_006464) with the red fluorescent protein (RFP) recombinant expression vectors were constructed using the pcDNA3.1(+)-RFP plasmid by GenScript (Nanjing, China) and named CALR-RFP and TGN38-RFP. All the resulting vectors were sequenced to verify their fidelity (compared with the target gene in cDNA) and orientation.

### 2.4. Cell Culture and Transfection

HEK293 cells were cultured at 37 °C in a humidified incubator with 5% CO_2_ in a high-sugar medium (Dulbecco’s Modified Eagle medium (DMEM), HyClone, Logan, UT, USA) supplemented with 10% fetal bovine serum (FBS, Hyclone, Waltham, MA, USA), 100 U/mL penicillin, and 100 μg/mL streptomycin (Procell, Wuhan, China). AjGPER1-EGFP, CALR-RFP, or TGN38-RFP vectors were transfected alone or cotransfected into HEK293 cells using X-tremeGENE HP DNA transfection reagent (Roche Applied Science, Indianapolis, IN, USA) as required. The cells were then incubated in 6-well plates for 24 h and inoculated into 24-well plates for subsequent experiments. To eliminate the effects of FBS, HEK293 cells were starved in FBS-free DMEM for 4 h prior to experiments.

### 2.5. Subcellular Localization of AjGPER1

To explore the subcellular localization of AjGPER1, the lipophilic cell-membrane probe DiI and endoplasmic reticulum or Golgi markers were used as subcellular markers in HEK293 cells. AjGPER1-EGFP was transfected alone or cotransfected with CALR-RFP or TGN38-RFP and incubated in 6-well plates. After 12–16 h of incubation, the transfected HEK293 cells were seeded onto 24-well plates with glass coverslips at the bottom and incubated at 37 °C for 24 h. The AjGPER1-EGFP-expressing HEK293 cells were stained with the living cell membrane red dye DiI (Beyotime, Shanghai, China) at 37 °C. Subsequently, the cells were fixed with 4% paraformaldehyde in phosphate-buffered saline (PBS) for 15–20 min, incubated with 4′,6-diamidino-2-phenylindole (DAPI; Beyotime, Shanghai, China) for 10 min to stain the cell nuclei, and washed twice with cold PBS. The cells were visualized by fluorescence microscopy using a Leica TCS SP5II laser scanning confocal microscope to observe the receptor localization using an HCX PL APO lambda blue 63 × 1.4 oil immersion lens.

### 2.6. Experimental Design of AjGPER1 Receptor Activity Determination

To determine the activity of the AjGPER1 receptor in HEK293 cells, blank cells without transfection, control cells transfected with the pEGFP-N1 empty plasmid (Aoqian Biotechnology, Hangzhou, China), and HEK293 cells transiently expressing AjGPER1-EGFP were seeded into 24-well plates and then starved in serum-free medium for 4 h to reduce the background activation of extracellular regulated protein kinases 1/2 (ERK1/2) and eliminate the effects of medium changes. The cells were then stimulated with the same concentrations of E_2_ or BPA (100 nM, diluted in serum-free medium). HEK293 cells expressing AjGPER1-EGFP were pretreated with DMSO or various inhibitors for 1 h, followed by time-gradient stimulation with E_2_ (100 nM), after which the medium was quickly discarded for protein extraction. The inhibitors used were PKA inhibitor H89 (MedChemExpress, Monmouth Junction, NJ, USA), PKC inhibitor Go 6983 (MedChemExpress, Monmouth Junction, NJ, USA), or G_αq_ protein inhibitor FR900359 (provided by Zhejiang University, Hangzhou, China).

### 2.7. Western Blot Assay

Cells were lysed with radioimmunoprecipitation (RIPA) assay buffer (Beyotime, Shanghai, China) containing a 1% protease-inhibitor cocktail (Sigma-Aldrich, St. Louis, MO, USA) for 1 h at 4 °C. The cell lysates were denatured at a high temperature (100 °C), and an equal magnitude of cell lysates was then subjected to vertical electrophoresis by 10–12% sodium dodecyl sulfate-polyacrylamide gel electrophoresis (SDS-PAGE) gel and transferred to polyvinylidene difluoride (PVDF) membranes. Membranes were blocked with 5% skim milk for 1 h and then incubated at 4 °C overnight respectively with rabbit antiphospho-ERK1/2 antibodies (1:2000, Cell Signaling Technology, Danvers, MA, USA, 4370S, p-ERK1/2) and rabbit anti-ERK1/2 antibodies (1:2000, Cell Signaling Technology, Danvers, MA, USA, 4695S; as an internal reference protein, total-ERK1/2). The specificity of antibody binding in heterologous systems has been demonstrated previously. Immunoreactive bands were detected using a horseradish peroxidase-conjugated goat antirabbit IgG antibody (Beyotime, Shanghai, China). Protein bands were detected using an enhanced chemiluminescent substrate (Beyotime, Shanghai, China, A0208), and the intensity of the immunoblots was visualized and quantified using a ChemiScope 6100 Chemiluminescent Imaging System (Clinx Science Instruments, Shanghai, China). The image grayscale was subsequently analyzed using ImageJ, and statistical significance was assessed by one-way analysis of variance (ANOVA) followed by Tukey’s multiple comparison test using GraphPad Prism (Version 8.2.1).

### 2.8. Real-Time Quantitative PCR (qRT-PCR)

For gene-expression analysis, qRT-polymerase chain reaction (PCR) was performed using cDNA products from various tissues of *A. japonicus*. Due to its consistent expression in various *A. japonicus* tissues, *β-tublin* was selected as an internal-control gene [33]. Based on the CDS region, gene-specific primers were designed for both control and *AjGPER1* (Appendix A). The primers were tested to ensure that a single discrete band was amplified without primer dimers or other secondary structures. The qRT-PCR assays were conducted using the SYBR PrimeScript™ RT reagent kit (TaKaRa, Kusatsu, Japan) following the manufacturer’s instructions using the ABI 7500 Software (Version 2.0.6) (Applied Biosystems, Foster City, CA, USA). The relative levels of gene expression were estimated using the 2^−ΔΔCt^ method and fold changes in *AjGPER1* expression were normalized to those of the internal control gene expression [34]. Differences between groups were analyzed using one-way ANOVA, followed by Tukey’s multiple comparison test using GraphPad Prism (Version 8.2.1). The significance level was set at *p* < 0.05.

### 2.9. Tissue Culture and Treatment

For in vitro experiments, the ovary and respiratory tree tissues were cut into small pieces of approximately 70 mm^3^ and cultured in a humidified incubator at 18 °C in PBS medium supplemented with 12.0, 0.32, 0.36, 0.6, and 2.4 g/L of NaCl, KCl, CaCl_2_, Na_2_SO_4_, and MgCl_2_, respectively [35]. To investigate the effect of E_2_ and BPA on ERK1/2 phosphorylation in vitro, 10 pM (2.72 ng/L), 1 nM (272.38 ng/L), and 100 nM (27.24 μg/L) concentrations of E_2_, and 10 pM (2.28 ng/L), 1 nM (228.29 ng/L), and 100 nM (22.83 μg/L) concentrations of BPA were added to the above culture medium for 30 min of stimulation, followed by immediate protein extraction and Western blotting. To determine the enzyme activity, the cultured ovary and respiratory tree tissues were exposed to 100 nM (27.24 μg/L) of E_2_ or 100 nM (22.83 μg/L) of BPA at 18 °C for 4 h, with three biological replicates in each group.

### 2.10. Enzymatic Activities Determination

To investigate the effects of E_2_ (27.24 μg/L) or BPA (22.83 μg/L) exposure on glucose metabolism and antioxidant activity in the ovary and respiratory tree tissues of sea cucumbers, we evaluated the activities of glucose-metabolism-related enzymes trehalase (THL) [36], phosphofructokinase (PFK), and pyruvate kinase (PK) [37,38], and the antioxidant enzyme catalase (CAT) [39]. Following the application of E_2_ or BPA to the two tissues, 0.1–0.8 g of each tissue was homogenized to make 10% homogenate. After immediately centrifuging the homogenate at 4 °C for 10 min at 2000 rpm/min, the protein concentration of the sample was determined using a bicinchoninic acid (BCA) protein concentration determination kit (enhanced) (Beyotime, Shanghai, China). The supernatant was used to determine the activities of THL, PFK, PK, superoxide dismutase (SOD), and CAT using commercial kits (Jiancheng Institute of Biotechnology, Nanjing, China) according to the manufacturer’s instructions.

### 2.11. Statistical Analysis

GraphPad Prism (version 8.2.1) was used for statistical analysis. Gray-scale analysis of ERK1/2 phosphorylation in blank cells, control cells, and AjGPER1-EGFP-expressing HEK293 cells and qRT-PCR results were analyzed using independent sample t-tests. Gray-scale analysis of the ERK1/2 phosphorylation inhibitory effects of H89, Go 6983, or FR900359 in E_2_ stimulated AjGPER1-expressing cells and enzyme activity assay results were analyzed by one-way ANOVA followed by Tukey’s multiple comparison tests. Probability values less than or equal to 0.05 were considered to be statistically significant and probability values less than or equal to 0.001 were considered to be extremely significant (* *p* < 0.05; ** *p* < 0.01; *** *p* < 0.001; **** *p* < 0.0001). The *p* values are indicated in the figure legends, except for the qRT-PCR results, where different lowercase letters indicate significant differences. Results were expressed as mean ± standard error (SE) and all experimental data were obtained from at least three independent experiments with similar results.

## 3. Results

### 3.1. Characterization of AjGPER1

The putative *AjGPER1* sequence cloned from the cDNA of *A. japonicus* was 1197 bp in length and encoded 398 amino acid residues. The predicted molecular weight of the protein was 45.39 kDa and its theoretical isoelectric point was 8.93. Transmembrane analysis of its protein sequence revealed that it contains seven transmembrane helices (Figure 1) and that the transmembrane helix domain 6 (TM6) had a highly conserved CW×P motif, which is consistent with the fundamental characteristics of the GPCR superfamily.

The secondary structure was predicted using Protter 1.0 (Figure 2A), and 41 potential post-translational modification sites were predicted using NetNGlyc 1.0 and NetPhos 3.1: five potential N-linked glycosylation sites and 36 phosphorylation sites at 10 tyrosine, 9 threonine, and 17 serine residues. Robetta was used to construct a three-dimensional model of the AjGPER1 structure, which showed the typical GPCR molecular structure of the seven transmembrane helix (7TM) domains (Figure 2B). Multiple sequence alignments of AjGPER1 and GPER1s from other species were performed to evaluate their homology. Conserved residues are highlighted in the alignment (Figure 2C), showing relatively low levels of similarity between AjGPER1 and other GPER1s (25.40 to 28.62% sequence identity) (Appendix A).

### 3.2. Phylogenetic Analysis of AjGPER1

To analyze the relationship between AjGPER1 and estrogen receptors (ERs) from other species, we constructed a phylogenetic tree by Mega 5.0 using the amino acid sequences of AjGPER1, 51 alternate ERs, and 12 Kisspeptin1 receptors obtained from GenBank using the ML method (Figure 3). This result indicates that the deduced AjGPER1 amino acid sequence was positioned alongside the GPER1 family within the phylogenetic tree and clustered together in the context of ERs and Kisspeptin1 receptor subfamilies.

### 3.3. Subcellular Localization of AjGPER1-EGFP in HEK293 Cells

To verify the subcellular localization of putative AjGPER1, an AjGPER1-EGFP plasmid was constructed and transiently expressed in HEK293 cells with or without CALR-RFP or TGN38-RFP. Confocal microscopy (Figure 4) showed that AjGPER1 was partially expressed in the plasma membrane (colocalized with DiI), endoplasmic reticulum membrane (colocalized with CALR-RFP), and Golgi membrane (colocalized with TGN38-RFP). This molecular characteristic of AjGPER1 is consistent with that of vertebrate GPER1s, indicating that recombinant expression of EGFP does not affect the functioning of this receptor.

### 3.4. Estradiol-Induced ERK1/2 Phosphorylation in AjGPER1-EGFP-Expressing HEK293 Cells

Figure 5A shows that after 10 min of E_2_ stimulation, a significant ERK1/2 phosphorylation signal was detected in AjGPER1-expressing cells but not in the blank or control cells. Furthermore, we conducted a time-course assessment of the effects of 100 nM E_2_ administration. ERK1/2 phosphorylation significantly increased in a time-dependent manner before 15 min and then decreased to near-basal levels by 45 min (Figure 5B). E_2_ has been suggested to activate AjGPER1, triggering a time-dependent mitogen-activated protein kinase (MAPK) cascade in AjGPER1-EGFP-expressing HEK293 cells.

### 3.5. AjGPER1 Is Activated by E_2_ and Signals through the G_αq_-Dependent MAPK Pathway in HEK293 Cells

Different inhibitors have been used to study intracellular signaling pathways triggered by AjGPER1. Compared with ERK1/2 phosphorylation without any inhibitors, the p-ERK1/2 signal was significantly reduced by pretreatment with the PKA-specific inhibitor H89 (Figure 6B) and completely blocked by pretreatment with the PKC-specific inhibitor Go 6983 (Figure 6C) or G_αq_ protein inhibitor FR900359 (Figure 6D). Overall, these results suggest that once stimulated by agonist E_2_, AjGPER1 can activate the G_αq_ protein and mediate the MAPK cascade through the G_αq_/PKC/ERK1/2 signaling pathway, as well as the PKA/ERK1/2 signaling pathways (Figure 6F).

### 3.6. Bisphenol A-Induced ERK1/2 Phosphorylation in AjGPER1-EGFP-Expressing HEK293 Cells

To further investigate whether BPA administration also activates AjGPER1, the AjGPER1-EGFP-expressing HEK293 cells were exposed to a certain dose of BPA (100 nM) for 10 min. ERK1/2 phosphorylation was detected only in AjGPER1-EGFP-expressing cells in the presence of BPA, but not in blank or control cells (Figure 7A). We determined the time-dependent effects of BPA on ERK1/2 phosphorylation in AjGPER1-EGFP-expressing HEK293 cells from 0 to 45 min. The ERK1/2 phosphorylation signal induced by BPA (100 nM) stimulation significantly increased after 5–15 min and recovered to basal levels after 30 min (Figure 7B). These results indicate that BPA can activate AjGPER1, similar to E_2_, thus triggering the intracellular MAPK cascade.

### 3.7. Effects of BPA Exposure on Sea Cucumber Ovarian Tissue with Abundant Expression of AjGPER1

To investigate the expression patterns of *AjGPER1* in adult *A. japonicus*, its transcripts in the nerve ring, intestine, muscle, tentacle, respiratory tree, and ovaries were quantified using qRT-PCR. Gene expression was normalized to the expression of the housekeeping gene *β-tubulin*. Figure 8A shows that the mRNA expression of *AjGPER1* was detectable in all the tested tissue samples. The expression level of *AjGPER1* was significantly higher in the ovary than in other tissues (*p* < 0.05), whereas that in the respiratory tree tissues was relatively low.

The effect of BPA exposure on ERK1/2 phosphorylation in cultured tissues in vitro was evaluated by Western blotting, using E_2_ exposure as a positive control. Total ERK1/2 protein was detectable in the tissues of the respiratory tree and ovary, but phosphorylated ERK1/2 was not detected in the respiratory tree tissue (with low expression of *AjGPER1*). In contrast, phosphorylated ERK1/2 was detected in the ovarian tissue (with abundant expression of *AjGPER1*), and both E_2_- and BPA-induced ERK1/2 phosphorylation in the ovarian tissue was observed in a dose-dependent manner (Figure 8B). These results indicate that BPA exposure activates MAPK cascades in the ovarian tissue of sea cucumbers and may have effects similar to those induced by E_2_.

To further assess the physiological effects of BPA exposure on sea cucumbers, THL, PFK, and PK activities were measured to evaluate the changes in glucose metabolism in the respiratory tree and ovaries of *A. japonicus*. No significant differences were observed in the enzymatic activities of THL, PFK, or PK in the respiratory trees treated with BPA or E_2_. In the ovary, THL activity was markedly enhanced under E_2_ or BPA exposure (*p* < 0.05), PFK activity was significantly upregulated under BPA exposure (*p* < 0.05), and PK activity did not differ significantly (*p* > 0.05) (Figure 8C–E). The activities of CAT (Figure 8F) and SOD (Appendix A) were also quantified in the respiratory tree or ovary to represent the potential changes induced by BPA or E_2_ in antioxidant capacity. However, the results did not differ significantly (*p* > 0.05). Overall, these results indicate that short-term exposure to BPA in vitro increased the activities of THL and PFK enzymes in *A. japonicus* but did not affect antioxidant defense at certain concentrations.

## 4. Discussion

Due to its low cost, light weight, corrosion resistance, and water/air tightness, BPA is widely used in various plastic products [40,41]. Recent studies have shown that BPA is present in the oceans and negatively affects marine organisms and ecosystems [42,43,44]. Additionally, growing evidence indicates that BPA can interact with estrogen receptors and affect endocrine function through estrogen receptor-dependent signaling pathways, owing to its phenolic structure [45,46]. In recent years, GPER1 has been widely studied as a potential mediator of rapid estrogen-induced responses in vertebrates [47,48,49]. Moreover, the estrogenic effects mediated by GPER1s are potential targets for the disruption of various environmental estrogens [50]. However, the direct effects of BPA on GPER1 in marine animals, particularly invertebrates, have rarely been studied. In this study, we aimed to identify the molecular characteristics of AjGPER1 from *A. japonicus* and to test the bioactivity of BPA on AjGPER1 in the target tissue of the ovary to explore the potential reproductive toxicity of BPA on echinoderms.

The *AjGPER1* candidate was selected and predicted as the best hit from the *A. japonicus* genome (Accession: PRJNA413998) and from our database of *A. japonicus* GPCRs [28,51]. Further molecular characterization and functional identification of AjGPER1 revealed a typical seven-transmembrane structure, similar to that of other types of GPCRs, containing 36 predicted phosphorylation sites thought to be involved in regulating protein trafficking, localization, and signal transduction [52]. As a member of the G protein-coupled receptor family, AjGPER1, similar to vertebrate GPER1, has highly conserved regulatory elements such as the CW×P motif located in the TM6 and is critical for GPCR activation [53,54]. AjGPER1 also has a highly conserved CW×P motif in TM6, which is critical for GPCR activation [53]. Phylogenetic analysis showed that AjGPER1 clustered with GPER1 from other species in one branch, which differed significantly from the nuclear estrogen receptor and kisspeptin1 receptors [15]. In this phylogenetic tree, we put these nuclear receptor-type ERs and GPCR-type KISS1Rs together as an outgroup to show the difference, and two ER-like receptors from NCBI (PIK58758.1 and PIK52234.1) were not added to the tree, considering their low homology to known ERs. Although AjGPER1 showed low similarity to known GPER1s from invertebrates, we further investigated AjGPER1 by molecular functional characterization.

The sea cucumber, *A. japonicus,* as a nonmodel organism, is challenging to study with respect to its metabolic activity and signal transduction. Considering the high transfection efficiency, rapid proliferation, and evolutionarily conserved proteins in cell-signaling pathways in model cell lines [55], and the wide use of HEK293 cell lines in cell-signaling studies in invertebrates [56,57,58], we chose to use transfected HEK293 cells to determine the molecular and functional characteristics of AjGPER1. The limitations of studying sea cucumber GPER1 outside of its evolutionary context, to some extent, were complemented by in vitro experimental data from cultured *A. japonicus* ovarian tissues highly expressing this gene. In this case, the molecular activity of AjGPER1 responding to E_2_ and BPA could be better demonstrated. However, systematic research is required to provide conclusive evidence, particularly for echinodermic cells.

In AjGPER1-EGFP-expressing HEK293 cells, the subcellular localization of AjGPER1-EGFP observed by fluorescence microscopy was similar to that of native GPER1 in human cells, which is mainly expressed in the plasma, mitochondria, and Golgi membranes [59]. GPER1 is a nongenomic receptor that exerts rapid signal transduction effects [15]. Our data demonstrated that despite the low homology of AjGPER1 to vertebrate GPER1, the former was efficiently activated by E_2_, triggering the intracellular signal transduction of the G_αq_/PLC/PKC/ERK1/2 and cAMP/PKA/ERK1/2 signaling pathways, which is consistent with the functional activities identified in vertebrates [15,25,26]. These results suggest that AjGPER1 shares a conserved signal transduction mode with higher animals, even though these signaling cascades were observed in HEK293 cells rather than in the cells of *A. japonicus*.

Previous studies have shown that BPA directly affects GPER1s. In mammals, low concentrations of BPA activate ERK1/2 phosphorylation in mouse GC-2 cells [16] and cAMP response element-binding protein phosphorylation in human seminoma cells (JKT-1) [27]. In teleosts, BPA activates *D. rerio* GPER1 and *Oryzias latipes* GPER1 induces MAPK cascades [60,61]. In our study, we found that low concentrations of BPA or E_2_ successfully activated AjGPER1, inducing ERK1/2 phosphorylation in AjGPER1-expressing HEK293 cells and activating ERK phosphorylation in the ovarian tissues of sea cucumbers expressing high levels of *AjGPER1*. Nuclear estrogen receptors, which mainly belong to transcription factors, are rarely reported to have MAPK cascade activation biofunctions [62] but can be activated by MAPK signals [63]. Our results from HEK293 cells and cultured ovarian tissues collectively indicate that GPER1 can be activated by low BPA concentrations in marine invertebrates, although there was no direct evidence from genetic experiments to confirm the involvement of AjGPER1 in ovarian ERK phosphorylation. Therefore, BPA may exert toxic effects on marine invertebrates by mimicking estrogenic effects, leading to inappropriate activation of estrogen-initiated signaling pathways.

Notably, BPA has effects on the reproduction teleosts, including inhibition of oocyte maturation and obstruction of gonadal function [60,64]. Similar to most sea cucumbers, *A. japonicus* (exhibiting separate sexes) is a broadcast spawner [65]. Thus, environmental pollutants have toxic effects on the organs and offspring. In this study, we observed high levels of AjGPER1 expression in ovarian tissues and an ERK1/2 phosphorylation signal was detected after BPA treatment in vitro, indicating that the effects of short-term BPA exposure on ovarian tissues were similar to those of E_2_. Therefore, we believe that GPER1 plays a role in this process; however, further verification by gene interference or knockdown is required.

In echinoderms lacking an acquired immune system, reactive oxygen species (ROS) are crucial for cellular defense, and excessive ROS can cause oxidative damage to cells and tissues [44,66]. BPA has been reported to increase ROS generation and significantly reduce SOD and CAT levels in the liver and pancreatic tissues of rats and mice after chronic exposure [67], whereas CAT levels are significantly reduced in carp after BPA exposure [68]. However, no significant signs of oxidative damage were observed in this study. GPERs play a critical role in metabolic control in higher vertebrates [24]. In invertebrates, E_2_ exposure influences carbohydrate metabolism in the mussel *Mytilus galloprovincialis* by activating glycolysis-related enzymes [37]. In the present study, both THL and PFK activities were significantly increased in the ovarian tissues of *A. japonicus* after BPA exposure. This is consistent with previous studies showing that BPA-containing compounds significantly increase PFK activity in the digestive tract of *M. galloprovincialis* [69].

## 5. Conclusions

This study aimed to identify the molecular characteristics of AjGPER1 from *A. japonicus* in HEK293 cells and test the bioactivities of E_2_ and BPA on AjGPER1 in the target tissue of the ovary to explore the potential reproductive toxicity of BPA in sea cucumbers, which may provide a basis for the conservation of sea cucumber resources. To our knowledge, this is the first study to identify and functionally characterize GPER1 in *A. japonicus*. We demonstrated that E_2_ stimulates this receptor and activates the G_αq_/PLC/PKC/ERK1/2 and PKA/ERK1/2 signaling pathways. Low concentrations of BPA successfully activated AjGPER1, inducing ERK1/2 phosphorylation in AjGPER1-expressing cells and ovarian tissues of *A. japonicus*, indicating that GPER1 can also be activated by low BPA concentrations in sea cucumbers. Therefore, BPA may exert toxic effects on sea cucumbers by mimicking estrogenic effects, leading to inappropriate activation of estrogen-initiated signaling pathways. We observed high levels of AjGPER1 expression in the ovarian tissue, and the ERK1/2 phosphorylation signal was detected after BPA treatment in vitro, indicating that BPA possesses estrogen-analog bioactivity by activating AjGPER1. Our study also demonstrated that short-term BPA exposure caused a significant increase in THL and PFK activity in the ovarian tissues of *A. japonicus*, indicating that BPA causes metabolic disruption in sea cucumbers. Our findings provide a scientific reference for the molecular mechanism of BPA in marine animals and a basis for the ecological risk assessment of BPA and the conservation of wild sea cucumber species.

## Figures and Tables

**Figure 1 biology-12-00798-f001:**
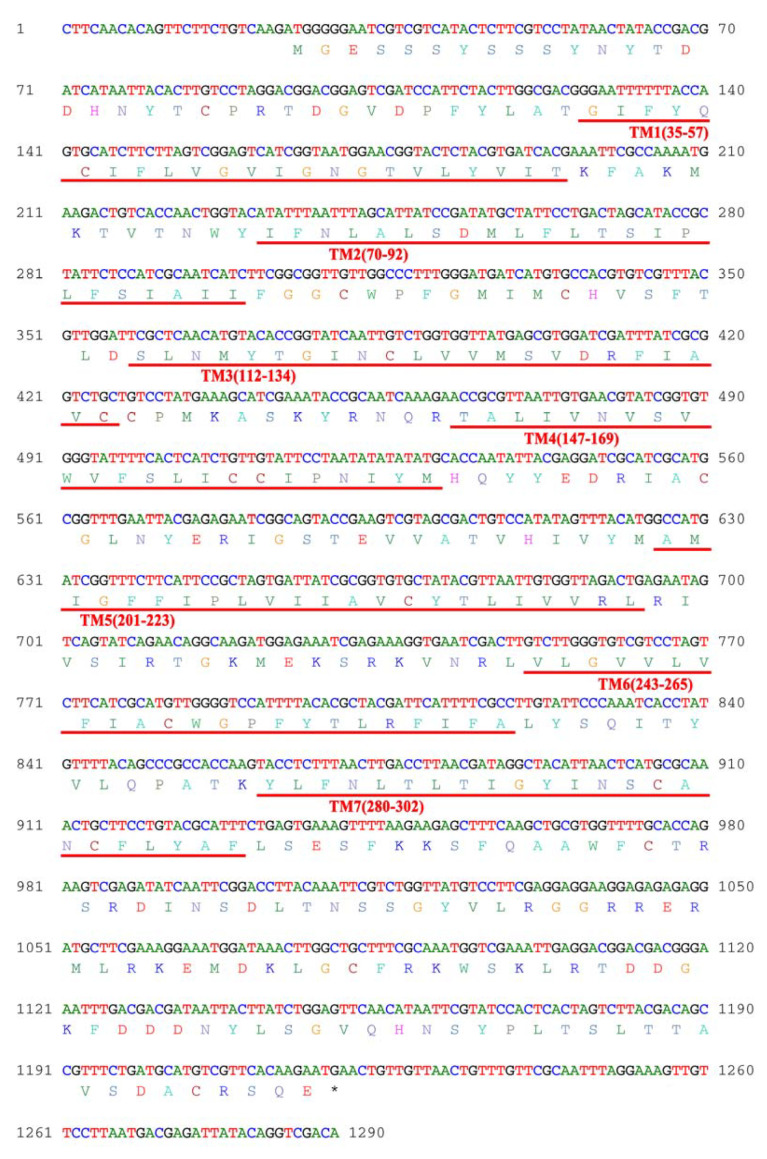
Coding sequence (CDS) and amino acid sequence of *AjGPER1*. Red underlines represent the seven transmembrane helices (7TMs).

**Figure 2 biology-12-00798-f002:**
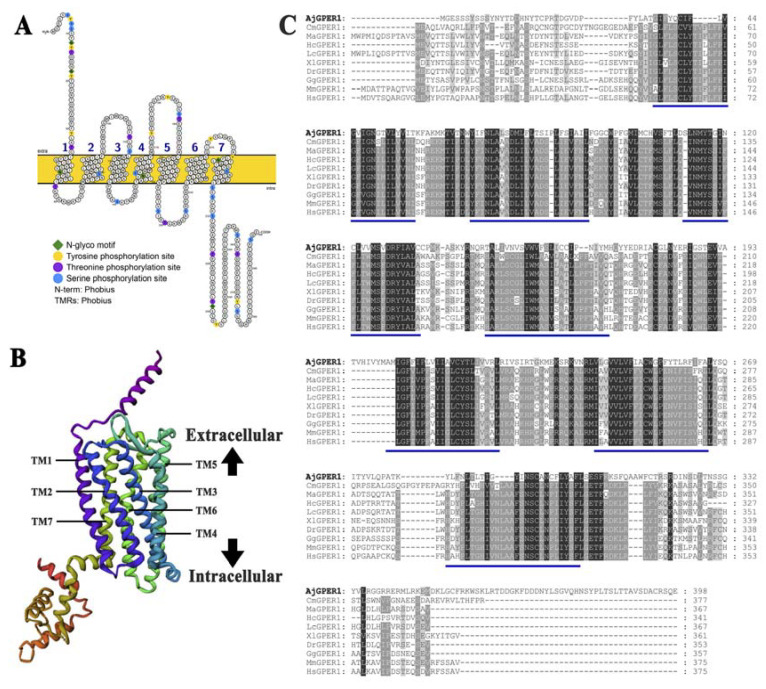
Structural characteristics and multiple sequence alignment of AjGPER1 amino acid sequences. (**A**) The 2D structure of the AjGPER1 amino acid sequences was predicated by Protter 1.0. N-glycosylation motifs were labeled with green diamonds, and tyrosine, threonine, and serine phosphorylation sites are indicated by yellow, purple, and blue circles. (**B**) The 3D structure of AjGPER1 protein was predicated using Robetta. (**C**) Multiple sequence alignment of GPER1s from *Apostichopus japonicus* and other vertebrate species. A black background represents the same amino acid residue. Sequences with a similarity greater than 80% are represented by a dark-gray background, and a light-gray background represents a similarity greater than 60%. The 7TMs are represented by the blue underlines. The GenBank accession numbers of *Callorhinchus milii* GPER1 (CmGPER1), *Monopterus albus* GPER1 (MaGPER1), *Hippocampus comes* GPER1 (HcGPER1), *Larimichthys crocea* GPER1 (LcGPER1), *Xenopus laevis* GPER1 (XlGPER1), *Danio rerio* GPER1 (DrGPER1), *Gallus gallus* GPER1 (GgGPER1), *Mus musculus* GPER1 (MmGPER1), and *Homo sapiens* GPER1 (HsGPER1) are presented in Appendix A.

**Figure 3 biology-12-00798-f003:**
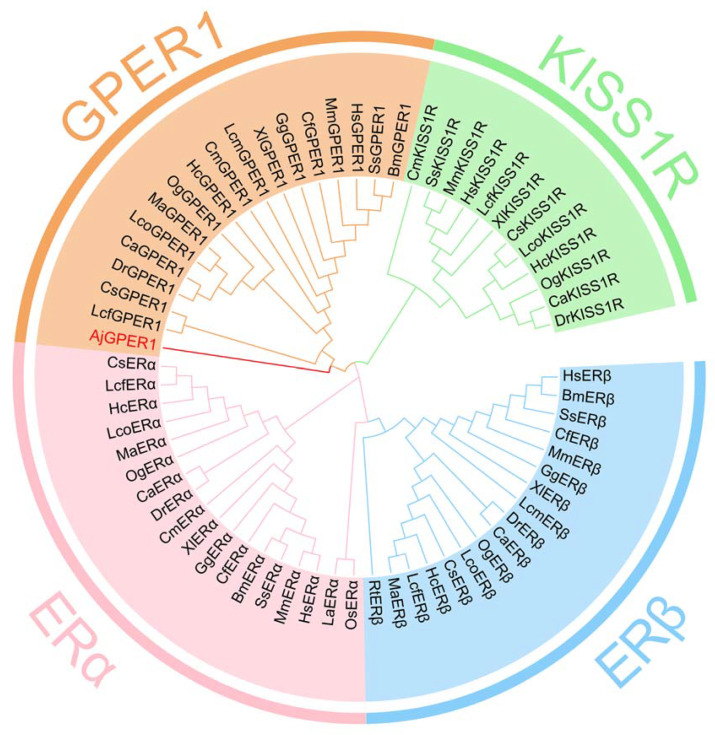
Phylogenetic analysis of GPER1 protein sequences. The phylogenetic tree was guided 1000 times with MEGA 5.0. The AjGPER1 was highlighted in red and the orange, pink, sky blue, or light green backgrounds represented the GPER1, ERα, ERβ, or KISS1R families, respectively. Sequence ID numbers of indicated receptors used in the phylogenetic analysis were elucidated in Appendix A.

**Figure 4 biology-12-00798-f004:**
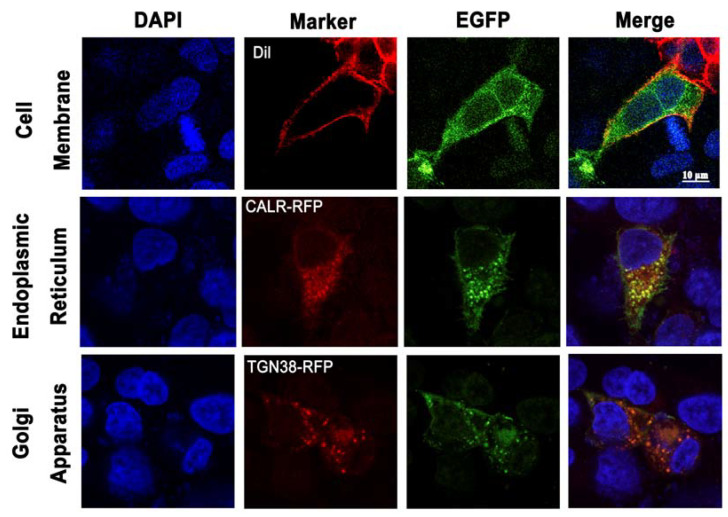
Localization of AjGPER1-EGFP fusion protein in HEK293 cells. Confocal microscopy of the HEK293 cells expressing AjGPER1-EGFP fusion protein. The nuclei and cell membrane were visualized with the nuclear probe (DAPI) and the cell membrane probe (DiI). The endoplasmic reticulum resident protein CALR and red-fluorescent protein fusion expression protein (CALR-RFP) or the Golgi body resident protein TGN38 and red-fluorescent protein fusion expression protein (TGN38-RFP) was coexpressed with AjGPER1-EGFP (scale = 10 μm). All images are representative of at least three independent experiments.

**Figure 5 biology-12-00798-f005:**
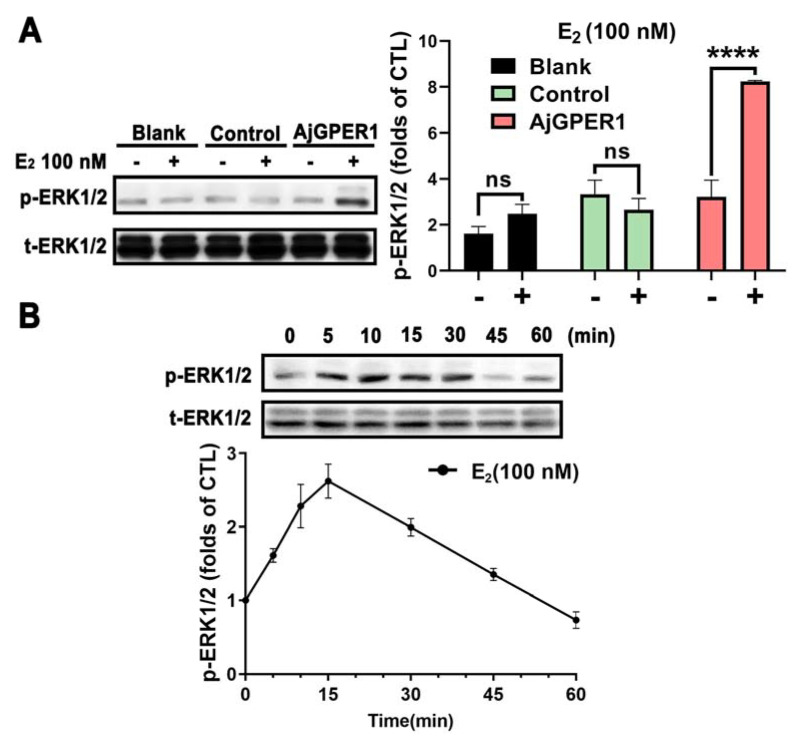
ERK1/2 phosphorylation in AjGPER1-EGFP-expressing HEK293 cells activated by estradiol (E_2_). (**A**) Blank (nontransfected) cells, control (transfected with empty vector pEGFP-N1) cells, and AjGPER1-EGFP-expressing cells were treated with DMSO or E_2_ (100 nM) for 10 min. (**B**) Time course of E_2_ (100 nM) treatment induced ERK1/2 phosphorylation in AjGPER1-EGFP-expressing HEK293 cells. p-ERK1/2 was normalized to the control (t-ERK1/2). Values are presented as means ± standard error of the mean (SEM) (n = 3). Data were analyzed using Student’s t-test (ns *p* > 0.05; **** *p* < 0.0001). All images and data shown are representative of at least three independent experiments. (See Appendix A for the original Western blot images.)

**Figure 6 biology-12-00798-f006:**
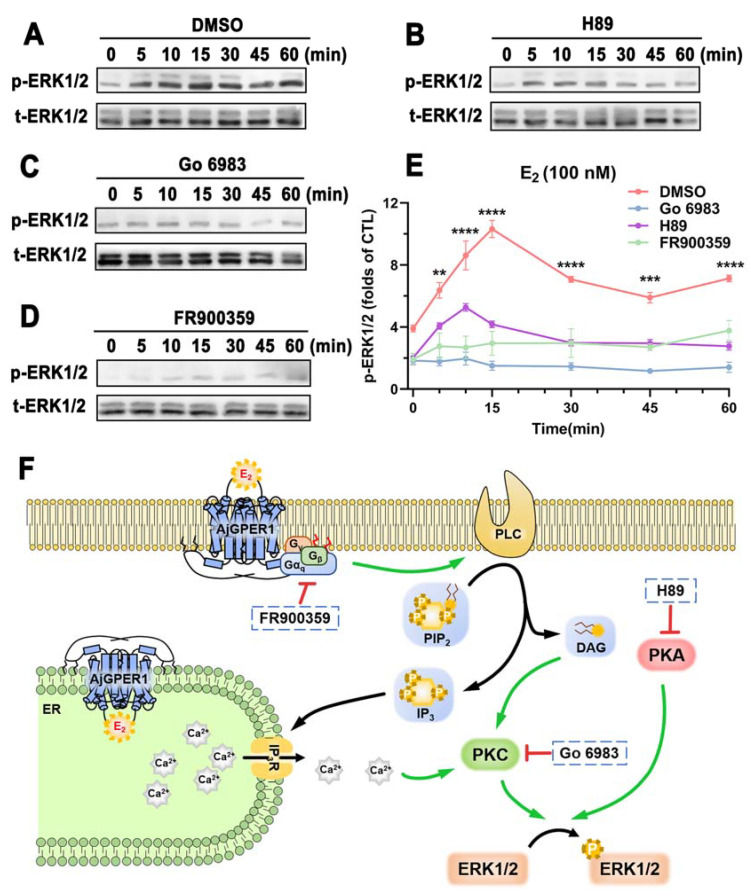
AjGPER1 activation-mediated intracellular signaling pathway. (**A**) AjGPER1 was activated by E_2_ (100 nM) to stimulate the phosphorylation of ERK1/2 in HEK293 cells. (**B**–**D**) Activation of ERK1/2 phosphorylation mediated by AjGPER1 by E_2_ was reduced or blocked by PKA, PKC, or G_αq_ protein inhibitors. Serum-starved AjGPER1-expressing cells were pretreated with DMSO, PKA inhibitor (H89, 10 μM), PKC inhibitor (Go 6983, 1 μM), or G_αq_ protein inhibitor (FR900359, 1 μM) for 1 h before E_2_ (100 nM) administration. (**E**) Statistical summary of the gray-scale analysis of ERK1/2 phosphorylation inhibitory effects by H89, Go 6983, or FR900359 in E_2_ stimulated AjGPER1-expressing cells. Values are presented as means ± SEM. Data were analyzed using one-way ANOVA followed by Tukey’s multiple comparison tests (** *p* < 0.01; *** *p* < 0.001; **** *p* < 0.0001). (**F**) Schematic representation of the AjGPER1-mediated cell signaling pathway. E_2_: estradiol; PLC: phospholipase C; PIP_2_: phosphatidylinositol (4,5) bisphosphate; DAG: diacylglycerol; IP_3_: inositol triphosphate; IP_3_R: inositol triphosphate receptor; ER: endoplasmic reticulum; Ca^2+^: calcium ion; PKC: protein kinase C; PKA: protein kinase A; ERK1/2: extracellular regulated kinase 1/2; P: phosphorylation. All pictures and data shown are representative of at least three independent experiments. (See Appendix A for the original Western blot images.)

**Figure 7 biology-12-00798-f007:**
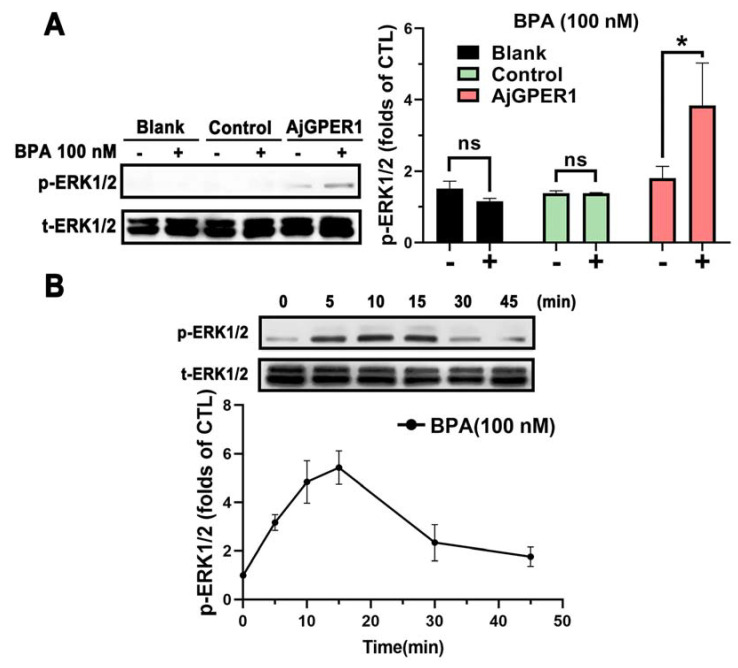
Activation of ERK1/2 in AjGPER1-expressing HEK293 cells by Bisphenol A (BPA). (**A**) Blank (nontransfected) cells, control (transfected with empty vector pEGFP-N1) cells, and AjGPER1-EGFP-expressing cells were stimulated with DMSO or BPA (10 μM) for 10 min. (**B**) Time course of BPA (100 nM)-induced ERK1/2 phosphorylation in AjGPER1-EGFP-expressing cells. The p-ERK1/2 was normalized to the loading control (t-ERK1/2). Values are represented as means ± SEM (n = 3). Data were analyzed using Student’s t-test (ns *p* > 0.05; * *p* < 0.05). All pictures and data shown are representative of at least three independent experiments. (See Appendix A for the original Western blot images.)

**Figure 8 biology-12-00798-f008:**
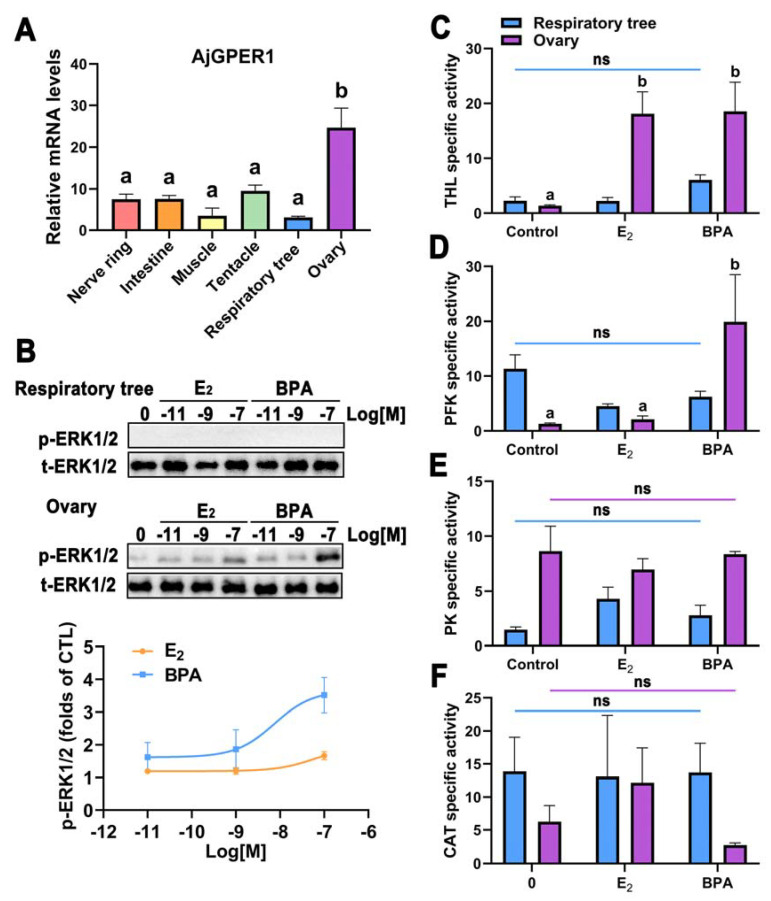
High expression of *AjGPER1* and physiological functions activated by E_2_ and BPA in sea cucumber ovarian tissues. (**A**) Tissue-specific expression profile of *AjGPER1* in multiple tissues. The relative expression level of AjGPER1 mRNA in nerve ring, intestine, muscle, tentacle, respiratory tree, and ovary. Data were presented as mean ± SEM (n = 4), and data were analyzed using an independent-sample t-test, and different lowercase letters indicated significant differences (*p* < 0.05) between tissues. (**B**) Activation of ERK1/2 in *A. japonicus* respiratory tree and ovary tissues by E_2_ and BPA. Tissues were stimulated with different concentrations of E_2_ or BPA for 30 min, and the p-ERK1/2 was normalized to the loading control (t-ERK1/2). Concentrations of E_2_ Log[M] −11: 2.7238 ng/L; −9: 272.382 ng/L; −7: 22.8286 μg/L and of BPA Log[M] −11: 2.2829 ng/L; −9: 228.286 ng/L; −7: 22.8286 μg/L. (**C**–**F**) Effects of 4 h exposure to E_2_ or BPA on enzyme activities in the respiratory tree and ovary tissue of *A. japonicus*. Values shown are multiples of the lowest activity of the control group expressed as means ± SEM (n = 3). Different color bars represent different tissues. Data were analyzed using one-way ANOVA followed by Tukey’s multiple comparison tests (ns *p* > 0.05), and different lowercase letters indicated significant differences (*p* < 0.05). All data shown are representative of at least three independent experiments. (See Appendix A for the original Western blot images.)

## Data Availability

The data supporting the findings of this study are available from the corresponding author upon reasonable request.

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
