# Peer review of "Bisphenol A (BPA) Directly Activates the G Protein-Coupled Estrogen Receptor 1 and Triggers the Metabolic Disruption in the Gonadal Tissue of Apostichopus japonicus"

_biology, 2023, doi:10.3390/biology12060798_

Round 1

Reviewer 1 Report

The manuscript entitled “Bisphenol A (BPA) directly activates the G protein-coupled estrogen receptor 1 and triggers the metabolic disruption in the gonadal tissue of Apostichopus japonicus” identifies the molecular characteristics of AjGPER1 from A. japonicus and tests the bioactivities of BPA on AjGPER1 in the target tissue of the ovary, then explores the potential reproductive toxicity of BPA on sea cucumber, which may provide a basis for the conservation of sea cucumber resources. Therefore, this research is valuable for physiological understanding and the conservation of sea cucumber resources. However, some questions need to be addressed before it is accepted by this journal.

1. Line 60, please write the full name of “DW”.

2. Line 76-77, the article describes that GPCR can serve as a regulator of estrogen induced physiological changes in vertebrates. The author should briefly introduce its regulatory mechanism or process as a reminder or reference for conducting this study.

3. Line 89, the adverbial phrases “in vitro” should be written in italic and other occurrences in the following contexts should be modified together.

4. Line 94, the word “identification” is suggested to be deleted.

5. Line 139, please revise the word “explpore”.

6. Line 157-159, in brief, what procedures were done to achieve the aim of HEK293 cells starvation in FBS-free DMEM for 4 h?

7. Line 180-186, I think description for steps here is too redundant, please optimize the description to make the experimental process simple and clear.

8. Figure 4, it seems less different in subcellular localization of AjGPER1 between cells transfected with CALR and TGN38 recombinant plasmids, for example, AjGPER1 is mostly located on cell membrane visualized by DiI, yet it is rarely exhibited on cell membrane in cell visualized by TGN38-RFP. Please explain it.

9. Line 321, the sentence “Estradiol-mediated ERK1/2 phosphorylation in AjGPER1-EGFP-expressing cells”, the word “Estradiol-mediated” is better to be replaced by “Estradiol-induced”.

10. Figure 5A, As I know, HEK293 cells should have background expression of GPER1, and under E2 stimulation, they should also be able to induce phosphorylation of ERK1/2. However, the positive signal in the figure did not increase under E2 stimulation in blank and control group. Please explain it.

11. Figure 5A and other figures for WB, the internal reference protein should be conducted to normalize protein content from different samples.

12. Line 342-345, the author needs to provide more evidence to obtain this conclusion. For example, some evidence that the suppressed genes can only participate in Gαq/PKC/ERK1/2 signaling pathway in vertebrates or invertebrates should be provided, otherwise, this conclusion will be inaccurate.

13. Figure 8C-F, please clarify that the results are from the samples stimulated by BPA or E2.

14. Figure 8B, why are the immunoblot of t-ERK1/2 in A. japonicus respiratory tree and ovary tissues different from those of HEK293 cell?

15. Line 441-443, and Line 447-448, the two sentences is repetitive, please revise.

16. Line 465,488,489,492, several species Latin names should be written in italic.

17. In reference, the formats of titles of articles are multifarious. For example, the first letter of some words in the middle sentence were uppercases. Please check and unitize.

Author Response

Response to Reviewer 1 Comments

Point 1: Line 60, please write the full name of “DW”.

Response 1: The full name “dry weight”of “DW” was added in revision.

Point 2: Line 76-77, the article describes that GPCR can serve as a regulator of estrogen induced physiological changes in vertebrates. The author should briefly introduce its regulatory mechanism or process as a reminder or reference for conducting this study.

Response 2: Brief introduction has been added to lines 78-81.

Point 3: Line 89, the adverbial phrases “in vitro” should be written in italic and other occurrences in the following contexts should be modified together.

Response 3: The font of these phrases has been corrected to italic.

Point 4: Line 94, the word “identification” is suggested to be deleted.

Response 4: The word “identification” has been deleted.

Point 5: Line 139, please revise the word “explpore”.

Response 5: The word has been corrected.

Point 6: Line 157-159, in brief, what procedures were done to achieve the aim of HEK293 cells starvation in FBS-free DMEM for 4 h?

Response 6: The FBS-free DMEM starvation were done to eliminate the influence of FBS on intracellular ERK1/2 phosphorylation in HEK293 cells.

Point 7: Line 180-186, I think description for steps here is too redundant, please optimize the description to make the experimental process simple and clear.

Response 7: The description of the experimental procedure has been optimized.

Point 8: Figure 4, it seems less different in subcellular localization of AjGPER1 between cells transfected with CALR and TGN38 recombinant plasmids, for example, AjGPER1 is mostly located on cell membrane visualized by DiI, yet it is rarely exhibited on cell membrane in cell visualized by TGN38-RFP. Please explain it.

Response 8: AjGPER1 has subcellular colocalization in both the cell membrane and the endoplasmic reticulum membrane, and Golgi membrane. There was also localization in the cell membrane in TGN38-RFP-labeled cells without DiI co-labeling, because the RFP and DiI are both red fluorescent.

Point 9: Line 321, the sentence “Estradiol-mediated ERK1/2 phosphorylation in AjGPER1-EGFP-expressing cells”, the word “Estradiol-mediated” is better to be replaced by “Estradiol-induced”.

Response 9: This has been modified.

Point 10: Figure 5A, As I know, HEK293 cells should have background expression of GPER1, and under E2 stimulation, they should also be able to induce phosphorylation of ERK1/2. However, the positive signal in the figure did not increase under E2 stimulation in blank and control group. Please explain it.

Response 10: HEK293 does not appear to have GPER1 expression. The gene expression profile of this cell line is linked below:

https://maayanlab.cloud/Harmonizome/gene_set/HEK293/BioGPS+Cell+Line+Gene+Expression+Profiles

The following article demonstrates that E2 in HEK293 has a negligible effect on G protein activation.

Thomas P, Pang Y, Filardo E J, et al. Identity of an estrogen membrane receptor coupled to a G protein in human breast cancer cells. Endocrinology, 2005, 146(2): 624–632.

Point 11: Figure 5A and other figures for WB, the internal reference protein should be conducted to normalize protein content from different samples.

Response 11: We used the total ERK protein as an internal reference protein to normalize the protein content of different samples. The instructions in the manuscript may not be clear enough. We have changed the statement in line 202-203. Here are some references that also use ERK as an internal reference protein:

Wang T, Cao Z, Shen Z, et al. Existence and functions of a kisspeptin neuropeptide signaling system in a non-chordate deuterostome species. eLife, 2020, 9: 1–28.

Feng J, Han T, Zhang Y, et al. Molecular characterization and biological function of CXCR1 in Nocardia seriolae-infected largemouth bass (Micropterus salmoides). Tissue and Cell, 2021, 72(April): 101551.

De Valdivia E G, Broselid S, Kahn R, et al. G protein-coupled estrogen receptor 1 (GPER1)/GPR30 increases ERK1/2 activity through PDZ motif-dependent and -independent mechanisms. Journal of Biological Chemistry, 2017, 292(24): 9932–9943.

Point 12: Line 342-345, the author needs to provide more evidence to obtain this conclusion. For example, some evidence that the suppressed genes can only participate in Gαq/PKC/ERK1/2 signaling pathway in vertebrates or invertebrates should be provided, otherwise, this conclusion will be inaccurate.

Response 12: In the discussion section, lines 476-478 have shown that GPER1 can mediate the Gαq/PLC/PKC/ERK signaling pathway in vertebrates. In this study, the pretreatment of HEK293 cells with Gαq protein inhibitor and PKC inhibitor showed that E2-induced the phosphorylation of ERK was inhibited. Therefore, we suggest that GPER1 mediates the Gαq/PKC/ERK signaling pathway.

Point 13: Figure 8C-F, please clarify that the results are from the samples stimulated by BPA or E2.

Response 13: BPA or E2 stimulation is explained below the horizontal coordinate in Figure 8C-F.

Point 14: Figure 8B, why are the immunoblot of t-ERK1/2 in A. japonicus respiratory tree and ovary tissues different from those of HEK293 cell?

Response 14: ERK proteins are species-specific, ERK also different from HEK293 cell in insect cell lines Sf21 in the following article.

Yang J, Huang H, Yang H, et al. Specific activation of the G protein-coupled receptor BNGR-A21 by the neuropeptide corazonin from the silkworm, Bombyx mori, dually couples to the Gq and Gs signaling cascades. Journal of Biological Chemistry, 2013, 288(17): 11662–11675.

Point 15: Line 441-443, and Line 447-448, the two sentences is repetitive, please revise.

Response 15: The sentences have been revised.

Point 16: Line 465,488,489,492, several species Latin names should be written in italic.

Response 16: The species Latin names have been checked and italicized.

Point 17: In reference, the formats of titles of articles are multifarious. For example, the first letter of some words in the middle sentence were uppercases. Please check and unitize.

Response 17: The reference format has been corrected.

Reviewer 2 Report

The research examined the effects of BPA on reproduction of sea cucumber by directive interaction to the estrogen receptor are appropriate for the journal. However, the MS has various weaknesses and need to be improved.

1.     There is insufficient information given about the study design and analysis in M&Ms. Such as, which data were analyzed using one-way ANOVA, which data were analyzed using independent t-test? Author should describe clearly in 2.10. Statistical analysis section.

2.     Lines 161-173: the content of this paragraph is exploring the subcellular localization of AjGPER1, so the title should be changed to be relevant its content.

3.     2.6 western blot assay: to determine the activity of the AjGPER1 receptor. So the content can be divided into two parts, one is the activity of the AjGPER1 receptor determination, the other part is western blot assay introduction.

4.     In 2.9, SOD was analyzed (line 242), but I can’t find the result about SOD in Results section.

5.     Author did not remark the meaning of “****” in Fig. 5A (line 355).

6.     Statistical analysis results should add in Fig. 5B.

7.     In fig. 6, author analyzed the data using the two-way ANOVA (lines 354-355), but I can’t find the result of two-way ANOVA. Author also didn’t introduce two-way ANONA in Statistical analysis.

8.     In lines 364-365: the AjGPER1-EGFP-expressing cells were exposed to a certain dose of BPA (100 nM) for 10 min, but the results (lines 369-370) and fig. 7B reflects that the exposure time more than 10 min. Please check and correct them.

9.     The Latin name of species should be italicized. (lines 465, 488, 492).

10.  In Discussion section, the first paragraph (lines 426-439) introduced the background and the purpose of the present study, which may be more suitable for Introduction section.

11.  The discussion section lacks discussion and analysis of their result with relevant research results. The authors need to improve the discussion focusing on the importance and the relevance of this study.

12.  Lines 493-500 made the conclusion of the research, but Conclusion section also concluded. Author should trim some content.

Author Response

Response to Reviewer 2 Comments

Point 1: There is insufficient information given about the study design and analysis in M&Ms. Such as, which data were analyzed using one-way ANOVA, which data were analyzed using independent t-test? Author should describe clearly in 2.10. Statistical analysis section.

Response 1: Thank you very much for pointing out our problems. A clearer description has been added.

Point 2: Lines 161-173: the content of this paragraph is exploring the subcellular localization of AjGPER1, so the title should be changed to be relevant its content.

Response 2: Title has been changed.

Point 3: 2.6 western blot assay: to determine the activity of the AjGPER1 receptor. So the content can be divided into two parts, one is the activity of the AjGPER1 receptor determination, the other part is western blot assay introduction.

Response 3: It has been revised.

Point 4: In 2.9, SOD was analyzed (line 242), but I can’t find the result about SOD in Results section.

Response 4: Since there was no significant change in SOD activity, the data was not shown in the article. We have added the results of SOD activity in the supplementary figure S1.

Point 5: Author did not remark the meaning of “****” in Fig. 5A (line 355).

Response 5: A description of the meaning of "****" has been added.

Point 6: Statistical analysis results should add in Fig. 5B.

Response 6: We havn’t performed statistical analysis and these line charts were only illustrated to show the trend of ERK1/2 phosphorylation over time.

Eg. The figure 6A-C in our previous paper: Wang T, Cao Z, Shen Z, et al. Existence and functions of a kisspeptin neuropeptide signaling system in a non-chordate deuterostome species. eLife, 2020, 9: 1–28.

Point 7: In fig. 6, author analyzed the data using the two-way ANOVA (lines 354-355), but I can’t find the result of two-way ANOVA. Author also didn’t introduce two-way ANONA in Statistical analysis.

Response 7: We are sorry for the writing mistake in this place and it has been corrected.

Point 8: In lines 364-365: the AjGPER1-EGFP-expressing cells were exposed to a certain dose of BPA (100 nM) for 10 min, but the results (lines 369-370) and fig. 7B reflects that the exposure time more than 10 min. Please check and correct them.

Response 8: Sorry for the confusing, the AjGPER1-EGFP-expressing cells exposing to a certain dose of BPA (100 nM) for 10 min is a description of Figure 7A. We have added a detailed description on line 382.

Point 9: The Latin name of species should be italicized. (lines 465, 488, 492).

Response 9: The Latin names of species have been italicized.

Point 10: In Discussion section, the first paragraph (lines 426-439) introduced the background and the purpose of the present study, which may be more suitable for Introduction section.

Response 10: Thanks for this comment. The first paragraph is a brief introduction about the backgroud and the purpose of the present study. So we believe that is is necessary for Discussion. Otherwise, these two apsects have been detailly described in Introduction section.

Point 11: The discussion section lacks discussion and analysis of their result with relevant research results. The authors need to improve the discussion focusing on the importance and the relevance of this study.

Response 11: The Discussion section has been reorgnized and we believe that in this version, the results have been discussed more clearly. The significance of this work has been concluded in Conclusion section.

Point 12: Lines 493-500 made the conclusion of the research, but Conclusion section also concluded. Author should trim some content.

Response 12: Sorry for the repetition. It has been modified.

Reviewer 3 Report

This paper studies the activation of a GPCR with sequence similarity to G protein coupled estrogen receptor 1 in human by Bisphenol A, an endocrine disrupting environmental pollutant. The results are interesting and important, but the following major changes are required before publication can be recommended.

The experimental evidence supporting the above conclusion is based on a mixture of studies with transfected A. japonicus AjGPER1 gene in human embryonic kidney cells (HEK293) and analysis of several tissues collected from A. japonicus animals. The results of the sea cucumber tissues and human cells are often described and discussed together, so that it is not always clear which is which. It is a very important difference if a result is obtained in HEK293 cells or sea cucumber tissue and keeping them apart is therefore of paramount importance.

A. Japonicus as a non-model organism is very challenging to study with respect to metabolic activity and signal transduction, and so the choice of using transfection in HEK293, while not ideal, is justified. However, this should be explicitly stated, and the discussion needs to have a clear paragraph about this exact point – the limitations of studying this gene out of its evolutionary context. Obviously, human and sea cucumbers have millions of years of evolution separating them and so it is not obvious that we can directly translate findings from HEK293 cells to sea cucumber tissues.

As far as I can see from the methods and results, the sources of materials are as follows (although this could be stated more clearly):

Figure 1-3 – bioinformatics

Figure 4 -7 - signaling-related Western blot analysis in HEK293 cells

Figure 8 – sea cucumber tissues

Thus, out of 8 figures, only 1 was actually data collected from the sea cucumber tissue. All other data (i.e. the vast majority) was inferred from predictions (bioinformatics) or studies in human cell culture. This is a major limitation of this paper that needs to be clearly stated and discussed.

The finding that the pERK1/2 antibody (I assume from human) recognizes a protein in A. japonicus that appears to show BPA stimulation, and to a lower extent E2 stimulation, is a major finding, since it is not obvious that the human antibody would recognize the A. japonicus protein. Since we have no proof that the band actually represents the ERK1/2 homologue – this also should be discussed. All we know is that there is cross-reactivity – it is likely that it is the ERK1/2 homologue but only an assumption until experimentally proven (future work?).

 The use of the term “similarity” in line 274 is incorrect – it should be sequence identity (as correctly stated in the supplementary table S3).

I would also replace “low levels” with “relatively low” levels. There should be a discussion section pointing out that this sequence identity means that there are other GPCR’s in A. japonicus that may be candidates for BPA/E2 binding. There was no analysis of all GPCR’s done in this paper that would prove that the AjGPER1 is truly the best hit compared to other GPCR’s in A. japonicus.

The phylogenetic analysis in Figure 3 is not very meaningful. GPCR’s are fundamentally different proteins than the gonadotropic receptors (GPCR transmembrane vs. transcription factor soluble). Obviously, AjGPER1 will group with the GPCR’s and not the GR’s. What exactly are the authors intending to show? An interesting question – which is not discussed in this paper, although it should be, is weather or not E2 and/or BPA might also binding to the transcription factors. Have the authors looked for GR homologues in the A. japonicus genome? I don’t see any Aj… labeled genes in the red and blue sections of the tree – if there are homologues, it would be nice to list them here. It is very possible, that the effects described in Figure 8 are actually mediated by the GR’s and not the GPCR, or both.

Figure 4 is not very helpful and should be moved to supplement. Transfecting a GPCR into HEK293 cells would obviously result in membrane targeting.

HEK293 cells should be stated in lines 364, 388 (?), 507 and wherever else the experiments were carried out in HEK293 cells.

Statatements such as “large amounts” should be substantiated with numbers. What concentrations?

The conclusion “These results suggest that AjGPER1 shares a conserved signal transduction mode with higher animals” is premature given that the only evidence is transfected gene in HEK293 cells that have the full apparatus of human signaling proteins. So far, the paper only shows that AjGPER1 can activate the human signaling cascade.

Author Response

Thanks again for the reviewer addressing the important issue in the manuscript. We are now able to respond to the comments brought up by reviewers in the revised manuscript. Enclosed please find our revised manuscript. Our responses are shown in "Blue" to round 1 revision, and "Red" to the round 2 for the convenience with the context. We hope that with these changes, the manuscript will now be acceptable for publication.

The detailed point to point responses are listed in attached file "Response to reviewer 3-MDPI R2".

Round 2

Reviewer 2 Report

The manuscript has been improved adequately in this version, and can be accepted for publication.

Author Response

Thanks again for your time and effort in evaluating our work. We do believe your suggestions vitally improved quality of the manuscript.

Reviewer 3 Report

Instead of only replying to my comments, the authors should make more extensive revisions in the text addressing my comments. I may not be the only one with these questions. 

Author Response

(The authors gave the same response as above.)

Round 3

Reviewer 3 Report

Point 5, the reference given in the response should be included in the manuscript along with the sentence "This AjGPER1 is the best hit from our database of A. japonicus GPCRs."

Point 6 should be added to the discussion

Author Response

Response to Reviewer 3 Comments

We appreciate the reviewer for her/his reviewing and the professional comments. We have corrected the indicated points to improve our manuscript, and thank the reviewer for her/his suggestions. Her/his comments gave us great help to optimize the MS.

Point 1: Point 5, the reference given in the response should be included in the manuscript along with the sentence "This AjGPER1 is the best hit from our database of A. japonicus GPCRs."

Response 1: Thanks again for your time and effort. We have added the sentence “The AjGPER1 candidate has been selected and predicted, as the best hit, from the A. japonicus genome (Accession: PRJNA413998), and our database of A. japonicus GPCRs” in green font (line 444-446) and added the following references:

Huang, D.; Zhang, B.; Han, T.; Liu, G.; Chen, X.; Zhao, Z.; Feng, J.; Yang, J.; Wang, T. Genome-wide prediction and comparative transcriptomic analysis reveals the G protein-coupled receptors involved in gonadal development of Apostichopus japonicus. Genomics 2021, 113, 967-978, doi:10.1016/j.ygeno.2020.10.030.

Point 2: Point 6 should be added to the discussion.

Response 2: We thank the reviewer for this suggestion. The indicated contents have been added into discussion section (line 455-458 and line 490-495) in green font. The inserted sentences are mainly about the discussion about the potential AjERs from A. japonicus and the functional involvment of nuclear estrogen receptors in ovary ERK1/2 phosphorylation under BPA and E2 stimulation.

In addition, we have made extensive English changes to the article (in green font), which may improve the language issues. Thanks for the comments.
